

WATER-USE DYNAMICS OF AN ALIEN INVADED RIPARIAN FOREST WITHIN THE
SUMMER RAINFALL ZONE OF SOUTH AFRICA
Scott-Shaw Bruce C[1] and Everson Colin S[1,2]
[1]Center for Water Resources Research, School of Agricultural, Earth and Environmental Sciences,
University of KwaZulu-Natal, Private Bag X01, Scottsville, Pietermaritzburg 3209, South Africa.
[2]Department of Plant and Soil Sciences, University of Pretoria, Private Bag X20, Hatfield, Pretoria
0028, South Africa.
**Abstract**
In South Africa the invasion of riparian forests by alien trees has the potential to affect the
country's limited water resources. Tree water-use measurements have therefore become an
important component of recent hydrological studies. It is difficult for South African
government initiatives, such as the Working for Water (WfW) alien clearing programme, to
justify alien tree removal and implement rehabilitation unless hydrological benefits are known.
Consequently water-use within a riparian forest in the upper Mgeni catchment of KwaZulu-
Natal in South Africa was monitored over a two year period. The site consisted of an indigenous
stand of eastern mistbelt forest that had been invaded by *Acacia mearnsii*, *Eucalyptus nitens*
and *Solanum mauritianum*. The heat ratio method of the heat pulse velocity sap flow technique
and the stem steady state techniques were used to measure the sap flow of a selection of
indigenous and introduced species. The indigenous trees at New Forest showed clear seasonal
trends in the daily sap flow rates varying from 8 to 25 L·day$^{-1}$ in summer (sap flow being
directly proportional to tree size). In the winter periods this was reduced to between 3 and 6
L·day$^{-1}$ when limited energy flux was available to drive the transpiration process. The water-
use in the *A. mearnsii* and *E. grandis* trees showed a slight seasonal trend, with a high flow
during the winter months in contrast to the indigenous species. The water-use in the understorey
indicated that multi-stemmed species used up to 12 L·day$^{-1}$. Small alien trees (<30 mm) *A.
mearnsii*, and *S. mauritianum* used up to 4 L·day$^{-1}$ each. The combined accumulated daily sap
flow per year for the three *A. mearnsii* and *E. grandis* trees was 6 548 and 7 405 L·year$^{-1}$
respectively. In contrast, the indigenous species averaged 2 934 L·year$^{-1}$, clearly demonstrating
the higher water-use of the introduced species. After spatial up-scaling, it was concluded that,
at the current state of invasion by 21 %, the stand used 40 % more water per unit area than if
the stand were in a pristine state. If the stand were to be heavily invaded, at the same stem
density of the indigenous forest, a 100 % increase in water-use would occur over an average
rainfall year.
*Key Words:    Indigenous trees, introduced trees, sap flow, transpiration, upscaling*



## 1. Introduction

Parts of South Africa experience up to 87% alien tree invasions (Working for Water, 2011), with most of these being in riparian areas that have readily available water and are difficult to manage. In South Africa there is a limited understanding of the extent to which tree species (particularly those in the riparian area) contribute to total evaporation. As such, it is difficult for government organizations and scientists to justify alien tree removal and rehabilitation unless a known hydrological benefit can be seen. The deep fertile soils, with high soil moisture contents associated with riparian areas, make them ideal for plant establishment and growth (Everson *et al*., 2007). In South Africa, these areas are extremely vulnerable to invasion by pioneer plant species, particularly species that have historically been introduced for commercial forestry. There is a widespread belief (which has been supported by numerous studies: Olbrich *et al*., 1996; Dye *et al*., 2001; Everson *et al*., 2007; Dye *et al*., 2008; Gush and Dye, 2008; Gush and Dye, 2009; Gush and Dye, 2015) in South Africa that indigenous tree species, in contrast to the introduced tree species, use less water and should be planted more widely in land rehabilitation programmes. Little research has been undertaken on the riparian area which excludes water limitations (except in severe drought conditions).

The benefits of healthy riparian zones in providing basic ecosystem services are well known (Askey-Dorin *et al*. 1999; Richardson *et al*., 2007). These benefits and the impacts of degradation through alien plant invasions were fully described in a study by Scott-Shaw *et al*. (2017) on the water use of plants in the Mediterranean climate of the Western Cape region of South Africa (Scott-Shaw *et al*., 2017). Here we summarize the most important aspects relevant to this study.

1. Commercial forestry has been blamed for increasing the green water (water lost by total evaporation) and decreasing the blue water (water in rivers and dams) in areas across South Africa (Jewitt, 2006). For these reasons, invasive alien plants, particularly introduced commercial trees, are considered to be a major threat to water resources and biodiversity.
2. There is a widespread belief in South Africa and globally that indigenous tree species, in contrast to the introduced trees, are water efficient and should be planted more widely in land restoration programmes. This is based on observations that indigenous trees are generally slow growing, and that growth and water-use are broadly linked (Everson *et al*., 2008; Gush, 2011).
3. At the ecosystem scale, studies indicate that invasive species use 189 % more water than indigenous dominated stands, particularly in tropical moist forests (Nosetto *et al*., 2005; Yepez *et al*., 2005; Fritzsche *et al*., 2006). In the high rainfall areas of South Africa, invasive alien plants growing in riparian areas are estimated to reduce annual streamflow by 523 x $10^6$ m$^3$ with a predicted annual reduction estimated to be as high as 1 314 x $10^6$ m$^3$ if allowed to reach a fully invaded state (Cullis *et al*., 2007).
4. Management of invaded riparian zones can result in hydrological gains disproportionately greater than the catchment area affected, with up to three times more streamflow yield than upslope areas (Scott and Lesch, 1996; Scott, 1999).
5. For many field and modelling applications, accurate estimates of total evaporation (ET) are required, but are often lacking. Sap flux density measurements give precise information on flow directions as well as spatial and temporal flow distribution (Vandegehuchte and Steppe, 2013). The heat pulse velocity (HPV) method is the most accurate of the available methods when compared against gravimetric methods (Steppe *et al*., 2010; Vandegehuchte and Steppe, 2013).



The New Forest site in KwaZulu-Natal, South Africa is part of a Working for Water (WfW)
clearing programme. The government-funded WfW programme clears catchment areas of
invasive alien plants with the aim of restoring hydrological functioning while also providing
poverty relief to local communities through job creation (Turpie *et al*., 2008). The aim of this
study was to quantify the potential hydrological benefit of the conversion of invaded stands to
more pristine stands for forest management practices. A detailed ecological study was
undertaken in conjunction with the two year hydrological study.
**2.    Methods**
An overview of the study site, sampling design and equipment implemented to carry out the
study has been provided in this Section. Details on the Heat Pulse Velocity (HPV) technique
has been documented in a previous paper (Scott-Shaw *et al*., 2017) and will not been repeated
here.
**2.1.    The Study Area**
The New Forest riparian area is located at latitude 29°28'30" S and longitude 29°52'48" E at
approximately 1760 m above sea level (Figure 1). The riparian area occurs along a tributary to
the upper Umgeni River, within Quaternary Catchment (QC) U20A. The New Forest riparian
area falls within the Eastern Mistbelt forest zone, which is dominated by *Leucosidea sericea,*
*Halleria lucida, Celtis africana* and *Afrocarpus falcatus*. The surrounding natural areas are
covered by Highland Sourveld (Acocks' 1988) or Drakensberg Foothill Moist Grasslands
(Mucina and Rutherford, 2006). The study site is typical of invasive alien plant (IAP) invasion,
whereby plantations have been grown in traditionally fire dominated grasslands and have
subsequently invaded the surrounding riparian areas. Eastern Mistbelt forests can be
characterised by cool, tall inland forests (Pooley, 2003). The mountain slopes of the area consist
of fractured dolerite dykes and basaltic outpourings (Crowson, 2008). The soils show evidence
of high precipitation and age with shallow unstructured soils occurring on the upper slopes, red
a-pedal soils on the midslope and soils with a underlying G-horizon dominating the low lying
areas.
Approximately 80% of the precipitation occurs in the summer months, which mostly consists
of orographically-induced and squall-line thunderstorms (Schulze, 1982). Interception from
mist makes a large contribution to the seasonal precipitation and determines the distribution of
the mistbelt forest. The long-term mean annual precipitation is between 941 and 1000 mm with
a distinct dry season from May to August. Average air temperatures range from 25.2 °C in the
summer to 16.9 °C in the winter, with the highest air temperatures occurring on the North-
facing slopes. Cool mountain winds occur at night with warm up-valley winds occurring during
the day (Crowson, 2008). Strong berg (westerly) winds are prevalent during August to
September and play a significant role in the spread of fire (Schulze, 1982).
New Forest farm is privately owned. The area south of the Umgeni tributary has been planted
with *Acacia mearnsii* and *Pinus patula* since the 1960s. The riparian area has since been heavily
invaded (> 20 %) with *A. mearnsii*, *Eucalyptus nitens* and *Solanum mauritianum*. Riparian
invasive alien tree clearing by WfW has been ongoing in the area.



## 2.2 Sampling Design

Five sites, each representing frequently occurring indigenous and introduced tree species, were instrumented for water-use monitoring. These trees included a size range of invasive *A. mearnsii* and *E. nitens* trees; a selection of common indigenous trees such as *Gymnosporia buxifolia*, *Celtis africana* and *Searsia pyroides* and a selection of trees growing in the understory (*S. mauritianum*, *A. mearnsii* and *Buddleja salviifolia*). The leaf area index (LAI) within this stand was 3.1 during the summer months with a reduction to 2.2 during the winter months due to the presence of deciduous species. There was little variation in LAI throughout the forest due to a uniform invasion by introduced trees and the disturbed nature of the indigenous species across the stand.

The trees within the riparian forest were in a disturbed state. The overall canopy height of the indigenous species was low, ranging from 4.1 to 8.3 meters. The invasive species were much taller, ranging from 13.1 to 16.6 meters. The physical characteristics of each monitored tree is provided in Table 1. There was variability between the stem moisture content and wood density between species, which can be explained by the different physical characteristics of the trees measured (variations in sap wood depth and active xylem concentration). A forest ecology study (Everson *et al.*, 2016) undertaken at New Forest compiled stem density measurements for re-growth forest, invaded riparian areas and on *S. mauritianum* dominated plots. The findings indicated that in the riparian forest, there was a density of 1 632 stems·ha$^{-1}$ invasive species with 6 090 stems·ha$^{-1}$ of indigenous species. In the *S. mauritianum* plots, there were 1 337 stems·ha$^{-1}$ of the invasive species, with 2 600 stems·ha$^{-1}$ of the remaining indigenous species.

## 2.3. Meteorological Station

A meteorological station was established on the 19$^{th}$ of September 2012 at New Forest farm in a nearby natural grassland, 250 m from the tree monitoring sites. Rainfall (TE525, Texas Electronics Inc., Dallas, Texas, USA), using a tipping bucket rain gauge was measured at a height of 1.2 meters from the ground. Air temperature and relative humidity (HMP45C, Vaisala Inc., Helsinki, Finland), solar irradiance (LI-200, LI-COR, Lincoln, Nebraska, USA), net radiation (NR-Lite, Kipp and Zonen, Delft, The Netherlands) wind speed and direction (Model 03002, R.M. Young, Traverse city, Michigan, USA) were all measured at a height of 2 m from the ground. These were measured at a 10 s interval and the appropriate statistical outputs were recorded every hour. A flat and uniform short grassland area which was regularly mowed was selected to meet the requirement for FAO 56 reference evaporation calculation.

## 2.4 Tree Water-use Measurements

A Heat Pulse Velocity (HPV) system using the heat ratio algorithm (Burgess, 2001) was set up to monitor long-term sap flow on all of the selected trees over a three year period. The instrumentation is described further by Clulow *et al*. (2013) and Scott-Shaw *et al*. (2017) and included hourly measurements of sap flow heater trace using a pair of type T-thermocouple probes. Regular maintenance was undertaken to ensure sufficient power and operation of the equipment. Measurements of sapwood depth, wood density and moisture content (described by Marshall, 1958) were taken to allow for up-scaling of probe measurements to whole tree water use (L·h$^{-1}$). Non-functional or damaged xylem (referred to as wounding) around the thermocouples was accounted for using wound correction coefficients described by Burgess (2001). Tree growth was recorded during each site visit by measuring diameter at breast height





and canopy height using a VL402 hypsometer (Haglöf, Sweden). Leaf area index using a LAI-
2200 (LI-COR, Lincoln, Nebraska, USA) was measured regularly throughout the stand. The
riparian forest had a limited aerodynamic fetch, which was not appropriate for the eddy
covariance and scintillometry techniques. Although the riparian stand had a heterogeneous
composition, the availability of detailed stem density measurements allowed for a methodology
to be followed based on recent up-scaling studies (Ford *et al*., 2007; Miller *et al*., 2007).
The Stem Steady State (SSS) technique, which estimates sap flow by solving a heat balance
for a segment of stem that is supplied with a known amount of heat (Grime and Sinclair, 1999),
was implemented on the smaller trees in the under-storey that were not quantifiable using the
HPV technique. Two Dynamax Flow 32-K systems (Dynamax, Houston, TX, USA) were
installed at New Forest. Each of these systems was powered by a 12V 100Ah battery, and
consisted of a CR1000 data logger (Campbell Scientific Inc.) and an AM16/32B multiplexer.
A voltage control unit regulated the voltage output depending on the number of collars and the
size of the collars. The gauge's insulating sheath (referred to as a 'collar') contains a system of
thermocouples that measure temperature gradients associated with conductive heat losses
vertically (up and down the stem), and radially through the sheath (Allen and Grime, 1994). A
foam insulation and weather shield were installed around the stem in order to sufficiently
minimize extraneous thermal gradients that could influence the heated section of stem (Smith
and Allen, 1996). The conduction of heat vertically upwards and downwards was calculated
by measuring voltages which corresponded to the temperature difference between two points
above and below the heater (Savage *et al*., 2000). The radial heat was calculated by measuring
the temperature difference of the insulated layer surrounding the heater (Savage *et al*., 2000).
Finally the voltage applied to the heater was measured. These measurements allowed the
energy flux (J.s$^{-1}$) to be calculated (Savage *et al*., 2000).
**2.5   Soil Water Measurements**
Hourly volumetric soil water contents were recorded at sites 1 and 2 within the riparian forest
with three time domain reflectometry (TDR) probes (Campbell Scientific Inc, CS 615) installed
horizontally at each site. The probes were installed at depths of 0.1, 0.3 and 0.5 m below the
litter layer, due to shallow soils at the site. A thick litter layer was observed throughout the site
consisting of mostly indigenous leaves and large broken branches from cattle and climatic
disturbances. The hourly volumetric water content measurements provided an understanding
of the responses of trees to rainfall events, or stressed conditions. Additional soil samples were
taken to determine the distribution of roots, soil bulk density and soil water content.
**3.   Results**
**3.1.   Weather Conditions during the Study Period**
The historical mean annual precipitation (MAP) for the New Forest area is 941 mm. During
the two-year monitoring period the area received 1164 and 1110 mm·a$^{-1}$ for 2013 and 2014
respectively. The rainfall distribution had a strong seasonal trend throughout the two years with
an exceptionally high amount of 120 mm·day$^{-1}$ in November 2014 (Figure 2). The daily solar
radiation peaked at 39 MJ·m$^{-2}$ following the same seasonal trend to that of the daily air average
temperatures.
During periods of high solar radiation, the water vapour pressure deficit was high and
correlated to peaks in transpiration rates. An average daily air temperature of 18.4 °C was



recorded at New Forest in the summer months. During these months, daily maximum air
temperatures occasionally exceeded 30 ºC. During the winter months, the air temperatures
averaged 11.7 ºC due to numerous days with low solar radiation. Periods of low solar radiation
correspond to overcast and/or rainfall periods and would likely result in little to no transpiration
occurring. The daily reference total evaporation ($ET_o$), derived from data captured on site,
averaged approximately 1 mm·day$^{-1}$ in the winter period to 5 mm·day$^{-1}$ during the summer
period. The monthly climate data illustrates the seasonal rainfall and air temperature trend
(Figure 3). The seasonal distribution of rainfall is important as it is during these periods of
water scarcity where the vegetative water-use becomes significant.
**3.2.   Tree Water-Use**
The radial heat pulse velocity of a *G. buxifolia* was measured over a short summer period
(Figure 4). The velocity of water moving through the tree was highest (up to 20 cm·h$^{-1}$) nearest
to the bark. Probes inserted deeper in the tree (> 15 mm) measured very little flow suggesting
that there was less active xylem at these depths, resulting in a small sapwood area. During the
winter period the radial heat pulse velocity of *A. mearnsii* had maximal flow 25 mm below the
bark (Figure 5). There was still flow occurring at a depth of 35 mm, indicating a much bigger
sapwood area than that of the indigenous tree. Furthermore, the sap velocity was high, (> 20
cm·h$^{-1}$) even during the dry winter period. These findings also indicated that correct probe
placement is essential in accurately representing the entire sapwood area of each tree.
Individual whole tree water-use showed a clear seasonal water-use trend for the semi-
deciduous and deciduous indigenous species (Figure 6). This was attributed to fewer daylight
hours and less heat units during the winter months resulting in reduced available energy;
therefore limiting the transpiration process. The daily water-use of *S. pyroides* averaged 8
L·day$^{-1}$ in summer compared to 3 L·day$^{-1}$ in winter, resulting in an accumulated total water use
of 1639 L·year$^{-1}$ (Figure 6 a). The deciduous *C. africana* used large amounts of water in the
summer, with an average of 25 L·day$^{-1}$. In the winter periods, after leaf fall, this species used
no water, resulting in a reduction of the total annual water-use (4307 L·year$^{-1}$). In contrast, *G.*
*buxifolia* used approximately 15 L·day$^{-1}$ in summer compared to 6 L·day$^{-1}$ in winter, resulting
in an accumulated total water use of 3870 L·year$^{-1}$ over the same period (Figure 6 a, b, c).
The introduced *A. mearnsii* of a similar stem diameter showed little seasonal variation (Figure
6 d). This tree averaged 22 L·day$^{-1}$ during summer periods and 14 L·day$^{-1}$ during winter periods
yielding a total of 5743 L·year$^{-1}$, higher than that of the indigenous species and comparable to
other large introduced species measured throughout South Africa (Gush *et al.*, 2015).
The water use of the multi-stemmed *B. salviifolia*, measured using the SSS technique, had the
highest daily water use (up to 12 L.day$^{-1}$) (Figure 7). This tree, although short, had the greatest
canopy area due to its lateral growth patterns with its numerous stems. In comparison, the
smaller *A. mearnsii* used considerably less water, with a peak of 4 L.day$^{-1}$. The three *S.*
*mauritianum* trees were highly variable, ranging from very low flows (0.4 L.day$^{-1}$) to in excess
of 4 L.day$^{-1}$. Although these values are small in comparison to the larger trees measured, it
does show the importance of the understorey in-stand measurements. These trees, particularly
the *S. mauritianum*, have a high density suggesting that the cumulative water-use of these trees
is important when scaling up to the total forest water use.





The daily summer water-use of indigenous trees at site 1 (Table 2) showed low water-use with an average of between 9 and 15 L·day$^{-1}$ in the summer months. Likewise, the indigenous trees at site 3 were low water users. Despite being deciduous, the *C. africana* used the most water of all the indigenous trees measured. This tree was the tallest of the indigenous trees measured and was not shaded by other species. Given that this species is deciduous, it is important to note that this tree uses a minimal amount of water in the winter when water resources are limited. The indigenous *B. salviifolia,* measured using the SSS technique had a similar water-use to that of the lower climax species.

The daily summer water-use of the *A. mearnsii* and the *E. grandis* were high in comparison to the indigenous species. These trees used between 18 and 27 L·day$^{-1}$ in the summer months and between 14 and 17 L·day$^{-1}$ in the winter months. On average, the introduced species used 2.4 times more water than the average indigenous species. However, this is a direct comparison and would differ to up-scaled comparisons due to the different stem densities of each species.

### 3.3. Soil Profile and Water Content

The volumetric soil water content (VWC) measured at New Forest was highly responsive to rainfall events (Figure 8 and 9). During the wet summer season, the VWC at the indigenous site 1 (Figure 6) ranged from 27 % in the upper horizon to 35 % in the lower horizon. This indicated a higher clay content in the lower horizon. Towards the dry season, as the vegetation continues to use water, the VWC was depleted to 10 % in the upper horizons. At the introduced site 2, the soils were uniform throughout the horizons. During the summer periods, the profile soil water averaged 27 % whereas it depleted to 9 % or 11 mm of water per 100 mm depth of soil during the dry periods.

The soils had a dry bulk density ($\rho b$) of 1.22 g.cm$^{-3}$, a particle density ($\rho_{particle}$) of 2.54 g.cm$^{-3}$ and a porosity 0.52, typically characteristic of sandy-loam soils. Introduced forestry species are known to have deep rooting systems, with observations of greater than 8 m in South Africa (Everson *et al*., 2006). This suggested that during dry periods, this stand can access water from deeper layers in the soil profile. However, given the shallow depth of all the soils and the close proximity of the sites to the stream, it is clear that the vegetation in this area was not limited by water availability.

The VWC at both sites did not respond significantly to rainfall events under 6 mm·h$^{-1}$ unless during consecutive events. Based on seasonally high transpiration rates we conclude that deep rooted plants in the riparian zone at the site are energy flux limited rather than moisture limited.

### 3.4. Upscaling Tree Water-Use

The results obtained from both the HPV and SSS techniques were used to determine an actual annual water-use per unit area of the invaded mistbelt forest. Two hypothetical scenarios, a pristine forest and a heavily invaded forest, were also tested. Using the stem density per size class taken from ecological research completed in the area (Everson *et al*., 2016), stands of forest were compared. As the forest did not have a closed canopy, understorey trees were numerous as more photosynthetically active radiation (PAR) was available throughout the stand. The water-use for a two-year average of the riparian forest in its current state (21 % invaded) was upscaled for all species and size classes. The total stand water-use was approximately 3.3 ML·ha$^{-1}$·a$^{-1}$ (330 mm·a$^{-1}$). This was 29 % of the average annual precipitation recorded during the monitoring period (1030 mm·a$^{-1}$).





Assuming that the site was rehabilitated to a more pristine state, using stem density for non-
invaded areas, the upscaled indigenous stand would use 2.39 ML·ha$^{-1}$·a$^{-1}$ (239 mm·a$^{-1}$). This
would be 21 % of the average annual precipitation. If the stand were to degrade further and
bcome heavily invaded, the upscaled invaded stand would use 4.88 ML·ha$^{-1}$·a$^{-1}$ (488 mm·a$^{-1}$).
This would be 43 % of the average annual precipitation. Based on these results we conclude
that the invaded stand uses 40 % more water per unit area annually than a pristine indigenous
stand. If the stand were to become heavily invaded, a two-fold increase in water-use would
occur (104 % increase) with concomitant impacts on the water balace (streamflow). The inter-
and intra-species water-use variations, particularly within the heterogenous indigenous stand,
highlight the importance of good replications of a representative sample tree species and size
classes. The results also show that it is important to highlight the slope position, physiological
characteristics and climatic variations occurring during measurement periods.
Due to a severe drought in this area, subsequent to the measurement period, these reults are
more likely to provide substance to land managers and decision makers, indicating the
hydrological benefit of restoration and rehabilitation activities.
**4.   Discussion and Conclusion**
In South Africa, it has been well documented that introduced commercial tree species, in
contrast to indigenous tree species, use more water and, if removed, would result in a net
hydrological gain (Olbrich *et al.*, 1996; Dye *et al.*, 2001; Everson *et al.*, 2007; Dye *et al.*, 2008;
Gush and Dye, 2008; Gush and Dye, 2009; Gush and Dye, 2015). The HPV and SSS techniques
have been used, both locally and internationally, on numerous vegetation types. The accuracy
of these measurements has been validated using gravimetric methods (Burgess *et al.*, 2001;
Granier and Loustau, 2001; O'Grady *et al.*, 2006; Steppe *et al.*, 2010; Vandegehuchte and
Steppe, 2013; Uddin and Smith, 2014; Forster, 2017). In South Africa, the HPV technique has
been shown to provide accurate estimates of sap flow in both introduced tree species such as
*Acacia mearnsii, Pinus patula* and *Eucalyptus nitens*, and indigenous tree species such as
*Rapanea melanophloeos, Podocarpus henkelii* and *Celtis africana* (Smith and Allen, 1996;
Dye *et al*, 2001; Everson *et al*, 2007; Dye *et al.*, 2008). There is consensus in the literature that
rehabilitation or restoration measures can result in maximising benefits such as goods and
services, while minimising water consumption (Gush, 2011).
A recent study, that was undertaken in conjunction with this study, showed that introduced
stands could use up to six times more water than indigenous species in the riparian area (Scott-
Shaw *et al.*, 2017). However, this difference was largely related to stem density at a site where
high winter rainfall and deep sandy soils were conducive to a high density mature introduced
stand. The stand at New Forest, which was highly disturbed and was in a constant state of
recovery, did not have a high stem density of mature trees in its current state. The measurements
undertaken at this site have allowed for an accurate direct comparison of indigenous and
introduced tree water-use. Additionally, the measurements of trees growing in the understorey
have provided interesting findings, indicating significant water-use in the sub-canopy layer.
The results showed that individual tree water-use is largely inter-species specific. As the
introduced species remain active during the dry winter periods, their cumulative water-use is
significantly greater than that of the indigenous species. Small tress (< 30 mm) in the
understorey can use up to 2000 L·year$^{-1}$, which is important for up-scaling to stand water-use.
Up-scaled comparisons showed that due to the invasion by *A. mearnsii* and *E. grandis* (21 %),
the stand water-use has increased by 40 %. This is an important finding as it provides clear





evidence to justify the hydrological benefit of clearing programmes. If the stand were to be
completely invaded, at the same stem density as the indigenous stand, the water-use would
double for this particular area. The findings from the understorey suggest that the water-use
from this zone should not be excluded from future studies, especially where there is no canopy
closure. The promotion of indigenous deciduous trees for rehabilitation or clearing programmes
may be important as there would be no transpiration during periods when water resources are
limited.
Spatial estimates of evapotranspiration are required but are difficult to obtain in remote areas
with limited aerodynamic reach. Remote sensing could be one area where this could be useful
given appropriate validation. However the nature of the "thin" riparian strip will require finer
scales than provided by most remote sensing products used for evaporation modelling (e.g.
Landsat 8). The use of drones could provide the best option for these narrow riparian strips in
the subsequent studies. Management dynamics are important in these environments. There is
potential for these data to be used in a modelling framework with specific inputs for invaded
mixed riparian forests. This would provide a suitable land management tool.

*Acknowledgements. The research presented in this paper forms part of an unsolicited research*
*project (Rehabilitation of alien invaded riparian zones and catchments using indigenous trees:*
*an assessment of indigenous tree water-use) that was initiated by the Water Research*
*Commission (WRC) of South Africa. The project was managed and funded by the WRC, with*
*co-funding and support provided by the Department of Economic Development, Tourism and*
*Environmental Affairs (EDTEA). The land owner, Alfie Messenger of New Forest farm is*
*acknowledged for allowing field work to be conducted on their property. Assistance in the field*
*by Dr. Alistair Clulow, Allister Starke and Siphiwe Mfeka is much appreciated.*





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



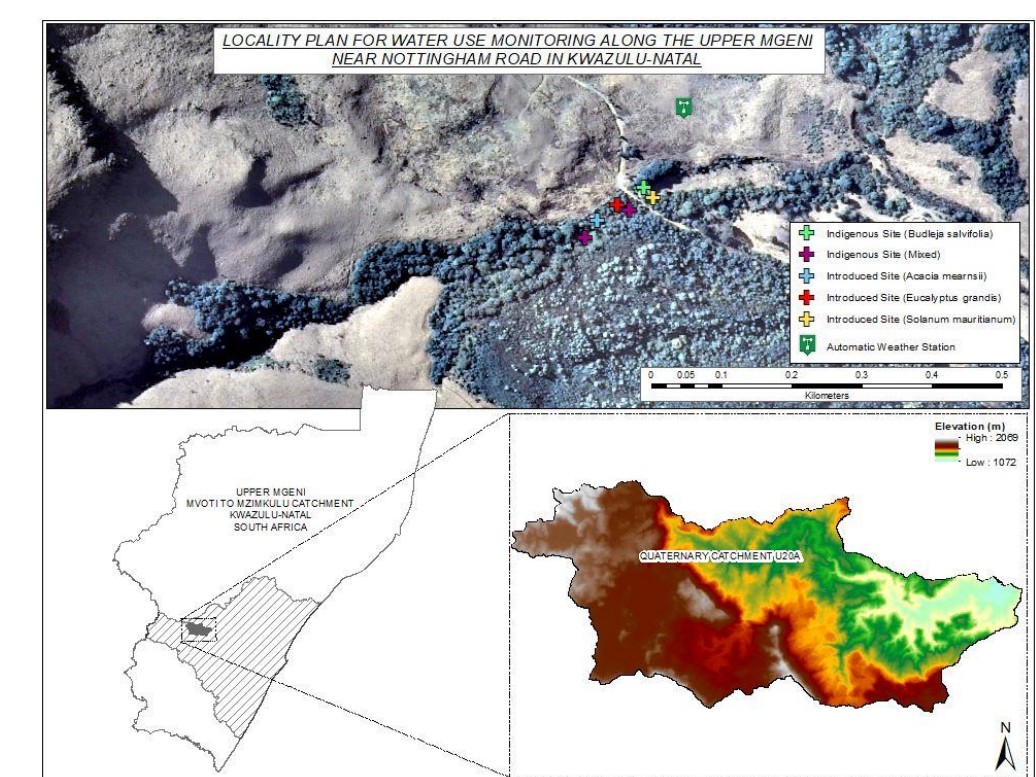

3          Figure 1.        Location of New Forest farm research area within KwaZulu-Natal, South Africa.



2    Table 1.    Tree physiology and specific data required for the calculation of sap flow and up-
3                                              scaling.

| Indigenous Forest (Site 1) | Diameter (mm) | Size Class | Moisture fraction | Average wounding (mm) | Wood density (kg·m⁻³) | Representative stem density (stems·ha⁻¹) |
|---|---|---|---|---|---|---|
| *Searsia pyroides* | 98 | Small | 0.41 | 3.1 | 0.60 | |
| *Gymnosporia buxifolia* | 114 | Small | 0.44 | 2.6 | 0.65 | 6 090 |
| *Gymnosporia buxifolia* | 58 | Small | 0.44 | 2.6 | 0.66 | |
| Introduced/Alien Forest (Site 2) | | | | | | |
| *Acacia mearnsii* | 131 | Medium | 0.48 | 3.0 | 0.69 | 1 632 |
| *Acacia mearnsii* | 166 | Medium | 0.47 | 3.0 | 0.69 | |
| Indigenous Forest (Site 3) | | | | | | |
| *Celtis africana* | 102 | Medium | 0.49 | 4.8 | 0.68 | |
| *Kiggerlaria africana* | 50 | Medium | 0.46 | 3.1 | 0.69 | 6 090 |
| *Leucosidea sericea* | 212 | Large | 0.47 | 2.8 | 0.64 | |
| Introduced/Alien Forest (Site 4) | | | | | | |
| *Eucalyptus nitenss* | 165 | Small | 0.51 | 3.8 | 0.71 | 1 632 |
| *Eucalyptus nitens* | 96 | Small | 0.51 | 3.9 | 0.71 | |
| Mixed understorey (Site 5) | | | | | | |
| #*Buddleja salvifolia* | 28⁺ | Small | N/A | N/A | N/A | 2 600 |
| #*Solanum mauritianum* | 25 | Small | N/A | N/A | N/A | - |
| #*Solanum mauritianum* | 10 | Small | N/A | N/A | N/A | - |
| #*Solanum mauritianum* | 19.1 | Small | N/A | N/A | N/A | 1 337 |
| #*Solanum mauritianum* | 26.7 | Small | N/A | N/A | N/A | - |
| #*Acacia mearnsii* | 25.6 | Small | N/A | N/A | N/A | - |

4    *Note: * indicates that the HPV technique was used and # indicates that the SSS technique was used. +indicates average stem diameter for multi-stemmed trees.





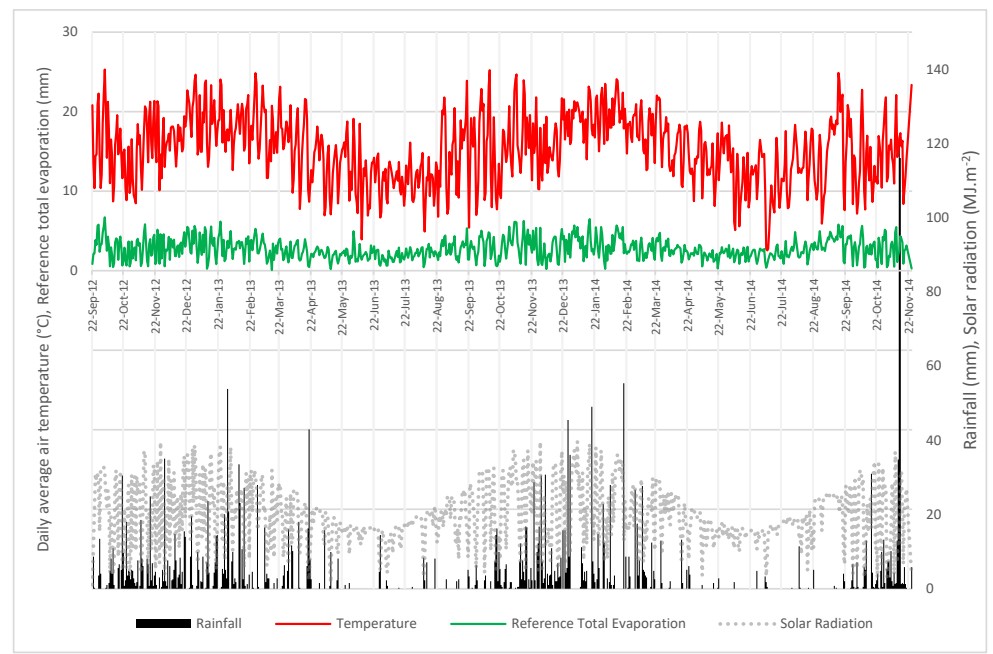

Figure 2.    The daily rainfall, solar radiation, average air temperatures and reference total evaporation at New Forest.

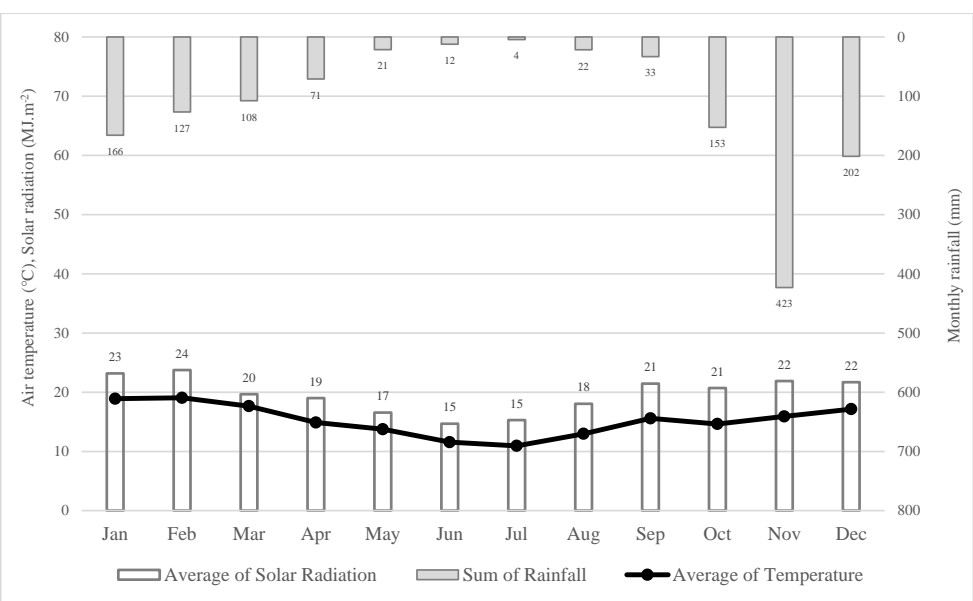

Figure 3.    The monthly rainfall, monthly solar radiant density, and average monthly air temperatures at New Forest averaged over two years.

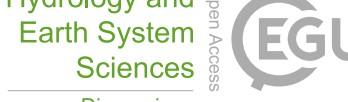



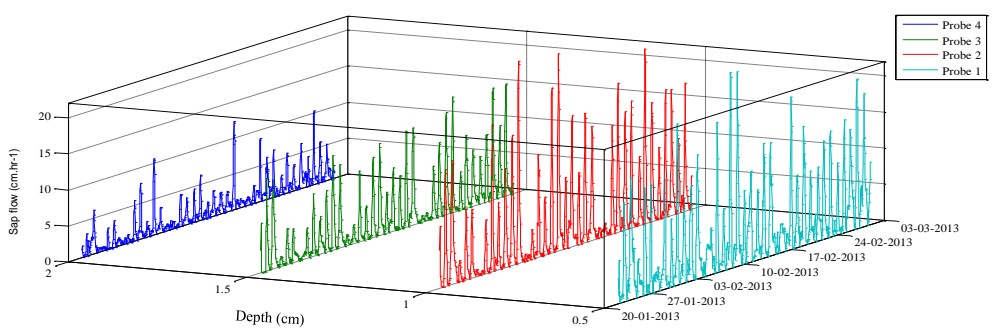

2        Figure 4.        Hourly heat pulse velocity of a *G. buxifolia* (Ø: 114 mm) at New Forest

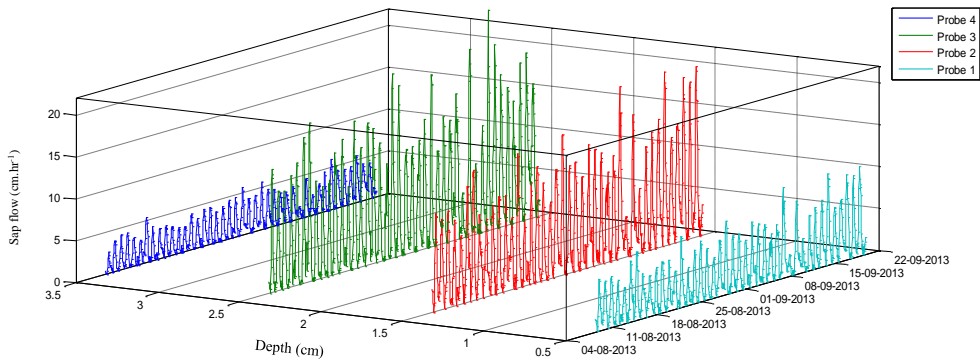

4        Figure 5.        Hourly heat pulse velocity of an *A. mearnsii* (Ø: 131 mm) at New Forest





5 Figure 6. Sap flow (daily and accumulated) averaged over two years (2013 & 2014) from an
6 indigenous S. pyroides (a), C. africana (b), G. buxifolia (c) and an introduced A.
7 mearnsii (d) at New Forest.



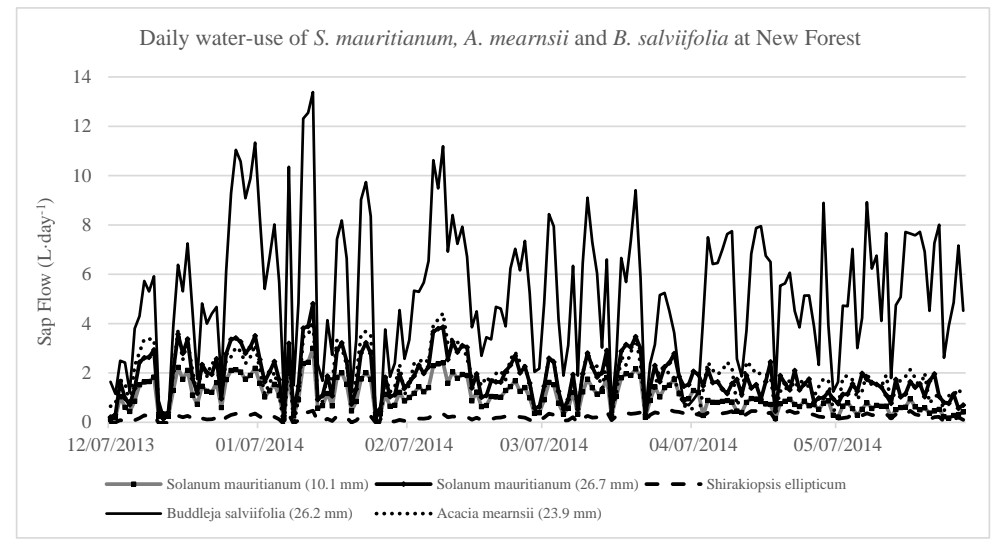

Figure 7.    Daily water-use for three S. mauritianum, a multi-stemmed B. salviifolia and an A.
mearnsii using the SSS technique at New Forest from December 2013 to June 2014.



2        Table 2.        Sap flow (daily and accumulated) for each species measured at New Forest

| Forest Type / Location | Species | Diameter (mm) | Daily Average Summer Sap Flow (L.d$^{-1}$) | Daily Average Winter Sap Flow (L.d$^{-1}$) | Annual Accumulated Sap Flow (L) |
|---|---|---|---|---|---|
| Indigenous Forest (Site 1) | *S. pyroides* | 98 | 9 | 3.6 | 1 639 |
| | *G. buxifolia* | 114 | 15 | 3.9 | 3 901 |
| | *G. buxifolia* | 58 | 12 | 3.8 | 2 883 |
| Introduced/Alien Forest (Site 2) | *A. mearnsii* | 131 | 18 | 15 | 5 786 |
| | *A. mearnsii* | 166 | 23 | 17 | 7 310 |
| Indigenous Forest (Site 3) | *C. africana* | 102 | 22 | 0.9 | 4 307 |
| | *K. africana* | 50 | 10 | 3.7 | 2 508 |
| | *L. sericea* | 212 | 9 | 4 | 2 369 |
| Introduced/Alien Forest (Site 4) | *E. grandis* | 165 | 27 | 15 | 7 668 |
| | *E. grandis* | 96 | 25 | 14 | 7 142 |
| Mixed understorey (Site 5) | *B. salviifolia* | 28 | 5.9 | 5.5 | 2 080 |
| | *S. mauritianum* | 25 | 0.4 | 0.3 | 127 |
| | *S. mauritianum* | 10 | 2.0 | 0.9 | 529 |
| | *S. mauritianum* | 19.1 | 2.9 | 1.2 | 748 |
| | *S. mauritianum* | 26.7 | 3.3 | 1.6 | 894 |
| | *A. mearnsii* | 25.6 | 3.4 | 1.8 | 949 |





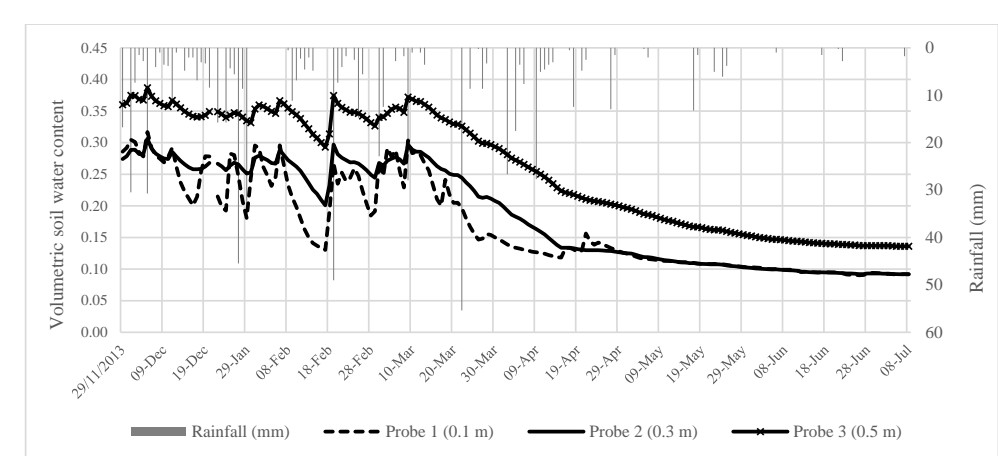

3    Figure 8.        Hourly volumetric soil water content and the hourly rainfall at site 1 at New Forest.

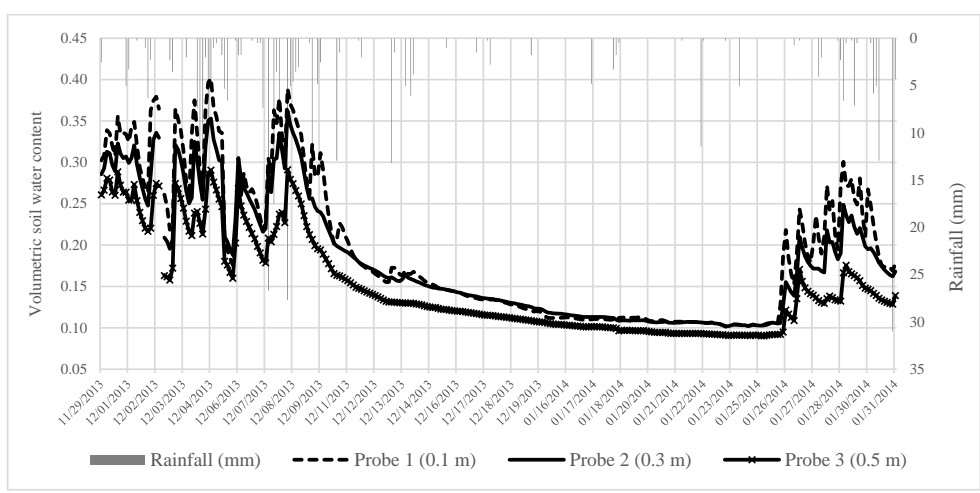

6    Figure 9.        Hourly soil volumetric water content and the hourly rainfall at site 2 at New Forest.
