# Peer review of "WATER-USE DYNAMICS OF AN ALIEN INVADED RIPARIAN FOREST WITHIN THE"

_Hydrology and Earth System Sciences, 2018_

## Referee Comment (RC1) · Anonymous Referee #1 · 2 Jul 2018

**GENERAL COMMENTS**

This study is a useful and needed contribution to knowledge about water use across different tree species and it is generally well written. Coming from a background of more ecosystem scale hydrology, I am not well qualified to comment on the details of the HPV and SS methods, and focus more the sampling design and scaling up of the results. The rationale for tree selection and methods of scaling up the numbers to the stand level requires more description than is currently given.

**SPECIFIC COMMENTS**

The description of the sampling design and its rationale needs more detail in the text.

[Figure]

Papers should be written to allow some level of replicability of the method to other sites. How and why were sites, species, and particular trees chosen? How did sites differ topographically, distance from river, soil properties, etc? Why were different numbers of trees of different species chosen? What is the likely or known age range or age structure in the indigenous and the invasive trees at the sites? Are the indigenous trees necessarily older? How did the species and the size classes of the trees compare to that of mature indigenous Eastern Mistbelt forest? Are any early successional? What is the typical composition of Eastern Mistbelt forest in terms of the proportion of trees that are deciduous? Deciduousness effects the water use of the trees. Was this proportion mimicked in the selection of trees to monitor? Were the Acacia and Eucalyptus trees near maturity?

Two of the three Acacia's and one of the two Eucalyptus trees measured had larger diameters than the indigenous trees, except for the L. sericea. Is statement on pg 7 ln 11 that "the introduced species used 2.4 times more water than the indigenous species" made by comparing individual trees of similar sizes or of similar ages?

The description of how the scaling up from individual tree water use to stand scale water use also needs more detail. Was the water use from different indigenous trees measured and applied across all trees of all species and size classes across the stand? Were water use figures of the individual measured trees applied to trees of the same species and/or functional group (e.g. deciduous or not, similar growth form or not, similar wood density) or similar size class? Wouldn't water use of mature trees be different to the young ones measured in this study? What was assumed to the be the size class distribution in the invaded and restored scenarios? The same as current or larger more mature trees assumed?

These aspects need to be described in the methods and the effects of the assumptions made, and alternatives, discussed in the discussions.

As such the figures of species level and stand level water use should also have some

estimates of likely uncertainty.

TECHNICAL CORRECTIONS

Pg 2, Ln 21-23 unnecessary to cite the paper twice in the sentence

Pg 3, Ln 35-37 The statement " invasive species use 189% more water than indigenous dominated stands" needs more clarification: this number is too specific to apply to all three of the cited studies. Was this the highest or lowest value from these three studies? Perhaps give the range of values across multiple studies. Does this only refer riparian forests compared to invaded stands?

Pg 4, Section 2.2 requires more details in the text as to the numbers of trees of different species and why they were chosen as well as how the scaling up calculations were done.

Pg 7 Ln 43 There is no citation for Everson et al 2016 in the reference list

pg 8 ln 4 typo: "bcome"

pg 15 Table 1 – typo: "Eucalyptus nitenss"

---

## Referee Comment (RC2) · T. Dube (Referee) · 3 Oct 2018

The study quantifies the potential hydrological benefit of the conversion of invaded stands to more pristine stands for forest management practices, in South Africa. The idea is scientifically novel and addresses key hydrological questions and the findings are likely to inform policy and decision making in the water sector. The paper needs great improvement before it can be published in HESS.

Comments. Title- The title requires rephrasing I failed to understand why the authors emphasize on the "SUMMER RAINFALL ZONE OF SOUTH AFRICA". Does this have anything to do with the spread of invasive or water use by these plants?

[Figure]

Abstract- general well written but I would recommend that authors include the objective of the study. As it is one has to speculate the direction of the study.

Introduction- This section is very weak and to general besides reading like a technical report. I would recommend that authors strengthen the motivation and support their argument with relevant literature. Authors should intensively interrogate literature and highlight scientific research strides that have been made as well as the gaps in knowledge that still need to be addressed. So far, this is totally missing. It is therefore very difficult for one to understand whether this is a technical report or a scientific study.

Methods – are poorly packaged and this makes it difficult for one to follow. I would, therefore, recommend that authors improve on this. The study area may is poorly drawn beside been illegible. A great improvement is required.

Results and discussion - although these sections read well they are very shallow and lack objectivity. The discussion is weak like the introduction; there is a lack of rigorous engagement of literature. Surprisingly there are too many references in the bibliography but the manuscript content does not demonstrate a thorough interrogation of literature.

Please also note the supplement to this comment:
https://www.hydrol-earth-syst-sci-discuss.net/hess-2018-227/hess-2018-227-RC2-supplement.pdf

---

## Referee Comment (RC4) · Anonymous Referee #3 · 17 Oct 2018

Review of Water-use dynamics of an alien invaded riparian forest within the summer rainfall zone of South Africa by Scot-Shaw and Everson.

The authors present a relevant data set on water use of several indigenous and invading tree species in the Mgeni catchment in KwaZulu-Natal in South Africa over a period of two years. The results show a remarkable difference in seasonal water use between indigenous and alien species. The alien species maintain high transpiration rates during the winter period, whereas the indigenous species strongly reduce transpiration during the winter months. Water use measurements taken at the level of individual trees were upscaled to the scale of a forest to quantitatively compare a pristine forest with an invaded forest scenario. The results may provide information for the management of this specific ecosystem.

Although the presented results appear sound and report relevant differences between native and alien trees species in this specific region, I'm under the impression that the authors are trying to slice their results a little too thin. The authors report that the hydrological campaign was conducted in conjunction with an ecological study (page 2, lines 6-7). However, the present study does not report on the ecological implications of the study. Are the ecological results to be reported in a separate study? Or is this the paper by Everson et al. (2016) that the authors refer to at a later stage in the manuscript? To me it remains unclear, specifically as I could not find the Everson et al. (2016) paper in the reference list. Despite the potential relevance of the species-specific water use measurements, the real added value of these data lies in their potential to indicate ecosystem benefits gained from removing or promoting the establishment of specific tree species relative to others. However, the current hydrological data set does not provide sufficient information to support such decisions. I therefore advise the authors to include data from their ecological study in the present hydrological study to interpret the hydrological differences between pristine and (heavily) invaded sites in terms of ecosystem functioning.

Comments: - Reverence list appears incomplete - Reported wood density in table 1 is in tonne mˆ-3, not kg mˆ-3 - Please report standard deviations alongside the averages in tables 1 and 2 to provide some information on the variability of the data underlying the average. - Please provide some more information on how the water use at the tree level was upscaled to the forest level. This is not described in the methods section at al. - Please analyze and discuss in more detail which plant functional traits determine the difference in water use between native and indigenous species. - Please indicate in figure 1 where exactly the site is located, perhaps by adding a dot in the lower right panel.

---

## Author Comment (AC1) · 5 Dec 2018

RC2 & RC3: 'Water-Use Dynamics of an Alien Invaded Riparian Forest Within the Summer Rainfall Zone of South Africa', Anonymous Referee #2 / T. Dube, 3 October 2018

HESS-2018-227

Anonymous referee #2 (AR2) is thanked for their thorough review. The thorough comments and suggestions provided were appreciated by the authors.

1. AR3 stated that the authors report that the hydrological campaign was conducted

in conjunction with an ecological study (page 2, lines 6-7). However, the present study does not report on the ecological implications of the study. Are the ecological results to be reported in a separate study? Or is this the paper by Everson et al. (2016) that the authors refer to at a later stage in the manuscript? • Although this study was undertaken in conjunction with an ecological study. Direct discussions with this component have been removed. Rather this paper is compared to a companion paper (hess-2016-650) that measured the water-use in a winter rainfall zone. The ecological findings are not reported on in this paper. However, more detail has been provided on the sampling strategy, density measurement and water-use up-scaling. The reference to Everson et al. (2016) has been updated in the reference list.

2. Despite the potential relevance of the species-specific water use measurements, the real added value of these data lies in their potential to indicate ecosystem benefits gained from removing or promoting the establishment of specific tree species relative to others. However, the current hydrological data set does not provide sufficient information to support such decisions. I therefore advise the authors to include data from their ecological study in the present hydrological study to interpret the hydrological differences between pristine and (heavily) invaded sites in terms of ecosystem functioning. • This comment, much like comment 1, requests more emphasis on the ecological component. The authors feel that providing detail on the ecological study would make paper too broad and require the inclusion of extensive literature, methods and results. This would detract from the quantitative findings provided in this study. As such, the authors have included only the necessary ecological methods and findings required to select the monitoring site, species and assist in up-scaling. The findings show a hydrological gain and not the changes in ecosystem functioning and other services.

Specific Comments

1. Reference list appears incomplete. • Checked and updated.

[Figure]

2. Reported wood density in table 1 is in tonne mËȨ-3, not kg mËȨ-3 • The values were corrected to g.cm-3 which is consistent with documented studies within South Africa and abroad.

3. Please report standard deviations alongside the averages in tables 1 and 2 to provide some information on the variability of the data underlying the average. • The standard deviations have been added to table 2 for each tree (calculated for the extent of the measurement period).

4. Please provide some more information on how the water use at the tree level was upscaled to the forest level. This is not described in the methods section at al. • A new Chapter (2.6) has been included detailing the up-scaling process. This Chapter also links to relevant Chapters on sampling design and species selection.

5. Please analyze and discuss in more detail which plant functional traits determine the difference in water use between native and indigenous species. • Although the functional traits that determine variations in water-use was not the focus of the study, it is an important component to discuss in this study. For example, "The greater sapwood area in introduced species, as well as their fast establishment, tree density and rapid growth, results in a greater transpiration rate than indigenous species per unit area." was included as a finding in this study.

6. Please indicate in figure 1 where exactly the site is located, perhaps by adding a dot in the lower right panel. • The location of the site relevant to the catchment and its elevation has been provided (yellow marker).

The study quantifies the potential hydrological benefit of the conversion of invaded stands to more pristine stands for forest management practices, in South Africa. The idea is scientifically novel and addresses key hydrological questions and the find-ings are likely to inform policy and decision making in the water sector. The paper needs great improvement before it can be published in HESS. Comments. Title- The title requires rephrasing I failed to understand why the authors emphasize on the "SUM-

MER RAINFALL ZONE OF SOUTH AFRICA". Does this have anything to do with the spread of invasive or water use by these plants?

Abstract-generalwellwrittenbutIwouldrecommendthatauthorsincludetheobjective of the study. As it is one has to speculate the direction of the study. Introduction- This section is very weak and to general besides reading like a technical report. I would recommend that authors strengthen the motivation and support their argument with relevant literature. Authors should intensively interrogate literature and highlight scientific research strides that have been made as well as the gaps in knowledge that still need to be addressed. So far, this is totally missing. It is therefore very difficult for one to understand whether this is a technical report or a scientific study. Methods – are poorly packaged and this makes it difficult for one to follow. I would, therefore, recommend that authors improve on this. The study area may is poorly drawn beside been illegible. A great improvement is required. Results and discussion - although these sections read well they are very shallow and lack objectivity. The discussion is weak like the introduction; there is a lack of rigorous engagement of literature. Surprisingly there are too many references in the bibliography but the manuscript content does not demonstrate a thorough interrogation of literature. Please also note the supplement to this comment: https://www.hydrol-earth-syst-sci-discuss.net/hess-2018-227/hess-2018-227-RC2supplement.pdf

The study quantifies the potential hydrological benefit of the conversion of invaded stands to more pristine stands for forest management practices, in South Africa. The idea is scientifically novel and addresses key hydrological questions and the findings are likely to inform policy and decision making in the water sector. The paper needs great improvement before it can be published in HESS. Comments. Title- The title requires rephrasing I failed to understand why the authors emphasize on the "SUMMER RAINFALL ZONE OF SOUTH AFRICA". Does this have anything to do with the spread of invasive or water use by these plants? Abstract-generalwellwrittenbutIwouldrecommendthatauthorsincludetheobjective

of the study. As it is one has to speculate the direction of the study. Introduction- This section is very weak and to general besides reading like a technical report. I would recommend that authors strengthen the motivation and support their argument with relevant literature. Authors should intensively interrogate literature and highlight scientïficresearch strides that have been made as well as the gaps in knowledge that still need to be addressed. So far, this is totally missing. It is therefore very difficult for one to understand whether this is a technical report or a scientïfic study. Methods – are poorly packaged and this makes it difficult for one to follow. I would, therefore, recommend that authors improve on this. The study area may is poorly drawn beside been illegible. A great improvement is required. Results and discussion - although these sections read well they are very shallow and lack objectivity. The discussion is weak like the introduction; there is a lack of rigorous engagement of literature. Surprisingly there are too many references in the bibliography but the manuscript content does not demonstrate a thorough interrogation of literature.

Please also note the supplement to this comment:
https://www.hydrol-earth-syst-sci-discuss.net/hess-2018-227/hess-2018-227-AC1-supplement.pdf

**Supplement:**

[revised manuscript text omitted]

---

## Author Comment (AC3) · 5 Dec 2018

RC1: 'Water-Use Dynamics of an Alien Invaded Riparian Forest Within the Summer Rainfall Zone of South Africa', Anonymous Referee #1, 2 July 2018

HESS-2018-227

Anonymous referee #1 (AR1) is thanked for their thorough review. The comments and suggestions provided were insightful and beneficial to the progress of this paper.

1. AR1 stated that the description of the sampling design and its rationale needs more detail in the text. Papers should be written to allow some level of replicability of the

method to other sites. The comments relating to up-scaling are itemised as follows:

• How and why were sites, species, and particular trees chosen? • This has been detailed in Chapter 2.2 which leads into the new Chapter 2.6.

• How did sites differ topographically, distance from river, soil properties, etc? • All sites were in close proximity to one another and the river. There were no variations in soils, climate and access to water. This has been mentioned in Chapter 2.2.

• Why were different numbers of trees of different species chosen? • This was due to availability of equipment and the associated budget constraints. Additionally, some trees which need to be in close proximity to the logger, do not provide good flow measurements. This limitation has been mentioned in Chapter 2.6.

• What is the likely or known age range or age structure in the indigenous and the invasive trees at the sites? • The growth stage of each tree has been included in Table 1.

• Are the indigenous trees necessarily older? • The growth stage of each tree has been included in Table 1. However, the authors were measuring the current state and therefore the water-use of the stand as it was during the measurement period. The selection of different size classes was more important than the age of each tree in this regard. Water-use does change with age but the size and LAI are more relevant to water-use. The state of the forest is addressed in the next comment.

• How did the species and the size classes of the trees compare to that of mature indigenous Eastern Mistbelt forest? • A description of the typical pristine composition and characteristics of Mistbelt forest has been provided in Chapter 2.1.

• Are any early successional? • The growth stage of each tree has been included in Table 1.

• What is the typical composition of Eastern Mistbelt forest in terms of the proportion of trees that are deciduous? • Provided in Chapter 2.1. The proportion of deciduous

species is variable in this forest type but the forest has been classified as Mistbelt forest.

• Deciduousness effects the water use of the trees. Was this proportion mimicked in the selection of trees to monitor? • Approximately 10% of species in the forest are deciduous, with the remainder being evergreen and semi-deciduous. The sampling design used this proportion in the selection of trees to monitor.

• Were the Acacia and Eucalyptus trees near maturity? • The growth stage of each tree has been included in Table 1.

2. Two of the three Acacia's and one of the two Eucalyptus trees measured had larger diameters than the indigenous trees, except for the L. sericea. Is statement on pg 7 ln 11 that "the introduced species used 2.4 times more water than the indigenous species" made by comparing individual trees of similar sizes or of similar ages? • This was a general statement based on an average water-use of each species. The size differences are noted and the results represent the status quo of how much water the stand was using during the measurement period. The statement provides an indication of what the invasion is using in comparison to typical indigenous trees that it has replaced.

3. The description of how the scaling up from individual tree water use to stand scale water use also needs more detail. Was the water use from different indigenous trees measured and applied across all trees of all species and size classes across the stand? • This reiteration of the previous comments has been addressed by Chapter 2.6 and links to the selection of trees and sampling design.

4. Were water use figures of the individual measured trees applied to trees of the same species and/or functional group (e.g. deciduous or not, similar growth form or not, similar wood density) or similar size class? • Yes, as described in the new Chapter 2.6, water-use was extrapolated per representative size class and species class identified in the density measurements.

5. Wouldn't water use of mature trees be different to the young ones measured in this study? • Yes, however the objective was to capture the age and size distribution of the stand and measure this as accurately as possible with the available equipment.

6. What was assumed to the be the size class distribution in the invaded and restored scenarios? The same as current or larger more mature trees assumed? • The hypothetical scenarios used the existing size class distribution for each species class and extrapolated this based on an assumed invasion level.

7. These aspects need to be described in the methods and the effects of the assumptions made, and alternatives, discussed in the discussions. As such the figures of species level and stand level water use should also have some estimates of likely uncertainty. • Comment no. 6 was described in the text as per this recommendation in Chapter 3.4. Due to the number of trees measured in each species class, the statistical level of uncertainty was not practical to include.

Specific Comments

1. Pg 2, Ln 21-23: unnecessary to cite the paper twice in the sentence. • Corrected.
2. Pg3, Ln 35-37: The statement "invasive species use 189% more water than indigenous dominated stands" needs more clarification: this number is too specific to apply to all three of the cited studies. Was this the highest or lowest value from these three studies? Perhaps give the range of values across multiple studies. Does this only refer riparian forests compared to invaded stands? • This was made more clear in the text. This was a global literature review of published studies. It provided a baseline from which the findings of this paper are compared to. "At the ecosystem scale, a comprehensive review of numerous internationally published studies indicate that invasive species use up to 189 % more water than indigenous dominated stands, particularly in tropical moist forests (Nosetto et al., 2005; Yepez et al., 2005; Fritzsche et al., 2006). These findings, typically outside of South Africa are limited to mostly herbaceous species with very few recent studies focusing on measurement of introduced

trees."

3. Pg4, Section 2.2 requires more details in the text as to the numbers of trees of different species and why they were chosen as well as how the scaling up calculations were done. • A new chapter in the Methods (2.6) has been added to provide detail on the up-scaling approach.

4. Pg 7 Ln 43 There is no citation for Everson et al 2016 in the reference list. • This has been added to the reference list.

5. Pg 8 ln 4 typo: "bcome". • Corrected.

6. Pg 15 Table 1 – typo: "Eucalyptus nitenss". • Corrected.

Please also note the supplement to this comment:
https://www.hydrol-earth-syst-sci-discuss.net/hess-2018-227/hess-2018-227-AC3-supplement.pdf

**Supplement:**

[revised manuscript text omitted]

---

## Author Comment (AC4) · 5 Dec 2018

RC2 & RC3: 'Water-Use Dynamics of an Alien Invaded Riparian Forest Within the Summer Rainfall Zone of South Africa', Anonymous Referee #2 / T. Dube, 3 October 2018

HESS-2018-227

Anonymous referee #2 (AR2) is thanked for their thorough review. The detailed comments and suggestions provided were appreciated by the authors.

1. AR3 stated that the title requires rephrasing. I failed to understand why the authors emphasize on the "SUMMER RAINFALL ZONE OF SOUTH AFRICA". Does this have anything to do with the spread of invasive or water use by these plants?   The authors used the term "summer rainfall zone" so that direct comparisons could be made to the companion paper (hess-2016-650) that measured the water-use in a winter rainfall zone. This term provides a broad climatic location, which is important when comparing the water-use to the climate of the study area. The inclusion of this statement prevents readers from taking the findings out of the climatic context. The authors have not changed the title as it would prevent the linkage to the companion paper.

2. Abstract-general well written but I would recommend that authors include the objective of the study. As it is one has to speculate the direction of the study.   This comment was noted and the following sentence was added to the abstract: The objective of this study was to investigate the water-use (transpiration rates) of a selection of introduced and indigenous tree species and quantify the hydrological benefit that could be achieved through a suitable rehabilitation programme.

3. Introduction- This section is very weak and to general besides reading like a technical report. I would recommend that authors strengthen the motivation and support their argument with relevant literature. Authors should intensively interrogate literature and highlight scientific research strides that have been made as well as the gaps in knowledge that still need to be addressed. So far, this is totally missing. It is therefore very difficult for one to understand whether this is a technical report or a scientific study.   A motivation was added to the introduction, discussing the problem, a potential solution and the reasoning behind the research approach. The authors have reviewed the literature citations in the paper and feel that it significantly covers the background of the study methods, the reason for undertaking the research and a comparison of the findings to documented findings. This should be considered in light of the fact that there is limited research on the riparian vegetation water-use, which the research findings should be compared to.

4. Methods – are poorly packaged and this makes it difiňĄcult for one to follow. I would, therefore, recommend that authors improve on this. • A new chapter was included detailing water-use up-scaling. Further comments on the methods from two other referees were addressed. Internationally accepted methods were not discussed in detail as it would be a repetition of documented literature.

5. The study area may is poorly drawn beside been illegible. A great improvement is required. • The authors feel that the study area is clearly legible. The location of the site within the catchment has been included.   6. Results and discussion - although these sections read well they are very shallow and lack objectivity. The discussion is weak like the introduction; there is a lack of rigorous engagement of literature. Surprisingly there are too many references in the bibliography but the manuscript content does not demonstrate a thorough interrogation of literature. • The authors appreciate the comment but feel that the context of the study has been provided (through an extensive literature review), the key findings are clearly discussed and future research provides a way forward for subsequent studies.

Please also note the supplement to this comment:
https://www.hydrol-earth-syst-sci-discuss.net/hess-2018-227/hess-2018-227-AC4-supplement.pdf

---

## Author Response (AR2)

**'Water-Use Dynamics of an Alien Invaded Riparian Forest within Summer Rainfall Zone of South Africa', Editor technical corrections, 21 January 2019**

**HESS-2018-227**

The editor, Prof. Dominic Mazvimavi, is thanked for his time, continued effort and comments provided throughout the development of this manuscript. The authors greatly appreciate his contributions. The details of the technical corrections are provided below (indicated in bold and italicised text):

1. **Page 1, line 1:** *The* taken out of title

2. **Page 1, line 33:** The *total accumulated* sap flow per year for the three *individual* A. mearnsii and E. grandis trees was 6 548 and 7 405 L·a$^{-1}$ respectively

3. **Page 1, line 36:** After spatial up-scaling, it was concluded that, at the current state of invasion (*21 % of the stand being alien species*), the stand used 40 % more water per unit area than if the stand were in a pristine state.

4. **Page 2, line 3:** *Ten million hectares of South Africa has been invaded by 180 alien species, which is over 8 percent of the country's total area* (van Wilgen *et al*., 2001). The majority of this invasion extent is within riparian areas that have readily available water and are difficult to manage (Working for Water, 2011).

5. **Page 2, line 8:** Can be *demonstrated*.

6. **Page 2, line 17:** Little research has been undertaken on the riparian area *where the availability of water to trees is often not limited* (except in severe drought conditions).

7. **Page 2, line 40:** In the high rainfall areas of South Africa, invasive alien plants growing in riparian areas (*5 726 km$^2$*) are estimated to reduce annual streamflow by 523 x 10$^6$ m$^3$ with a predicted annual reduction estimated to be as high as 1 314 x 10$^6$ m$^3$ if allowed to reach a fully invaded state (Cullis et al., 2007).

8. **Page 3, line 6:** This gap *in* knowledge has led to uncertainty and contention over riparian restoration and rehabilitation techniques.

9. **Page 3, line 19**: An overview of the study site, sampling design and equipment *used* to carry out the study has been provided in this Section.

10. **Page 3, line 29**: This forest type *when* in a pristine state…

11. *Page 3, line 31:* Mistbelt forest is a species rich, multi-layered *and has* a dense…

12. **Page 3, line 38:** Eastern Mistbelt forests can be characterised *as* cool, tall inland forests

13. **Page 3, line 41:** red a-pedal soils on the midslope*,* and soils

14. **Page 3, line 44:** Approximately 80% of the precipitation occurs in the summer months (*October to March*)…

15. **Page 3, line 47:** The long-term mean annual precipitation is between 941 and 1000 mm·*a$^{-1}$*

16. **Page 3, line 49:** with the highest air temperatures occurring on the *n*orth-facing slopes

17. **Page 4, line 2:** Strong berg (westerly) winds are prevalent during August to September and play a significant role in the spread of fire*s*

18. **Page 4, line 31:** The findings indicated that in the riparian forest, there was a density of 1 632 stems·ha$^{-1}$ *of* invasive species

19. **Page 4, line 38:** Rainfall using a tipping bucket rain gauge (TE525, Texas Electronics Inc., Dallas, Texas, USA). *Reference moved to the end of the sentence.*

20. **Page 5, line 7:** and *to* allow for

21. **Page 6, line 10:** Qstand. *Described in previous sentence as the stand water flux*.

22. **Page 6, line 41**: The seasonal distribution of rainfall is important as it is during these periods of water scarcity *when* the vegetative water-use becomes significant.

23. **Page 7, line 8:** This was attributed to fewer daylight hours and less heat units during the winter months *than in the summer months* resulting in

24. **Page 7, line 22:** *With regards to the understorey, the* water-use of the…

25. **Page 8, line 47:** *Due to the prevalence of severe droughts in this area,* these reults are more likely to provide substance to land managers and decision makers, indicating the hydrological benefit of restoration and rehabilitation activities. *Sentence reworded to avoid reference to recent drought.*

26. **Page 17, line 5:** *Added - N/A indicates that the measurement parameter was 'not applicable' for the technique used.*

27. **Page 20, line 5:** Figure 6. *Daily sap flow (dotted line) and accumulated sap flow (dashed line)* averaged over two years (2013 & 2014) from an indigenous *S. pyroides* (a), *C. africana* (b), *G. buxifolia* (c) and an introduced *A. mearnsii* (d) at New Forest.

28. **Page 21, line 1:** *Figure 7 legend changed to italics*

29. **Page 22, line 2**: Table 2 - Annual Accumulated Sap Flow (L·$a^{-1}$)

30. **Page 23:** *Figure 8 and 9, rainfall bar chart changed to blue and made more prominent*